# Structural Characterization, Rheological Properties and Protection of Oxidative Damage of an Exopolysaccharide from *Leuconostoc citreum* 1.2461 Fermented in Soybean Whey

**DOI:** 10.3390/foods11152283

**Published:** 2022-07-30

**Authors:** Yingying Li, Luyao Xiao, Juanjuan Tian, Xiaomeng Wang, Xueliang Zhang, Yong Fang, Wei Li

**Affiliations:** 1College of Food Science and Technology, Nanjing Agricultural University, Nanjing 210095, China; 2016108033@njau.edu.cn (Y.L.); 2021208015@njau.edu.cn (L.X.); 2018208012@njau.edu.cn (J.T.); 2019208028@njau.edu.cn (X.W.); 2020208016@njau.edu.cn (X.Z.); 2College of Food Science and Engineering, Nanjing University of Finance and Economic, Nanjing 210023, China

**Keywords:** *Leuconostoc citreum* 1.2461, soybean whey wastewater (SWW), exopolysaccharide (EPS) structural characterization, rheological properties, oxidative damage protection

## Abstract

Soybean whey is a kind of agricultural by-product enriched with nutritional value but with low utilization. The extracellular polysaccharides secreted by lactic acid bacteria during the fermentation possess a variety of structural characteristics and beneficial properties. In this study, an exopolysaccharide (EPS) was isolated from *Leuconostoc citreum* 1.2461 after fermentation in optimized soybean whey-enriched 10% sucrose at 37 °C for 24 h. The water-soluble EPS-1 was obtained by DEAE-52 anion exchange chromatography, and the structural characterization of EPS-1 was investigated. The EPS-1 was homogeneous with an average molecular weight of 4.712 × 10^6^ Da and consisted mainly of glucose. Nuclear magnetic resonance (NMR) spectrum and flourier transform infrared (FT-IR) spectrum indicated that the EPS-1 contained →3)-α-D-Glcp-(1→ and →6)-α-D-Glcp-(1→ residues. The rheological properties of EPS-1 under the conditions of changing shear rate, concentration, temperature and coexisting ions showed its pseudoplastic fluid behaviors. In addition, the EPS-1 exhibited certain scavenging activity on the ABTS radical and chelating activity on metal ions at relatively high concentrations. Furthermore, EPS-1 with a certain concentration was confirmed to have significant protective effects on yeast cell injury induced by hydrogen peroxide. This study reported the structural characteristics of exopolysaccharide from *Lc. citreum* 1.2461 and provides a basis for its potential application in the field of functional foods.

## 1. Introduction

Polysaccharides from animals, plants and varieties of microorganisms have been proven to exhibit functional and physicochemical properties, e.g., their antioxidant and immunological activity, rheological properties, water retention ability and filming capacity [1,2]. Currently, water-soluble polysaccharides have attracted increasing interest owing to their biological activities and potential processing properties [3]. Compared with the plant- and algae-derived water-soluble polysaccharides, exopolysaccharides (EPSs) from microorganisms occupy only a small part of the market. The bioactivity of EPS has been linked to their specific structure, including monosaccharide composition, molecular weight (Mw) and linkage pattern, as well as culture conditions [4,5]. EPS obtained from lactic acid bacteria (LAB) are generally composed of a single type of sugar subunit (homopolysaccharide) or consist of several types of sugar subunits (heteropolysaccharide). As a typical homopolysaccharide and hydrocolloid, dextran and its derivatives have attracted great attention for its industrial and medical applications [6]. *Leuconostoc* species are the primary producers of dextran with different structures and probiotic properties. According to previous research, the EPS from *Lc. citreum* B-2 was confirmed to be a glucan with a highly branched structure and showed certain antioxidant potential [7]. Another EPS derived from *L.citreum* NM105 was also characterized to be a water-soluble polysaccharide with α-(1→6) linkage in the backbone with α-(1→2) linkage in the branch point [8].

In general, water, inorganic salt, carbohydrate and nitrogen source are essential nutrients for the growth of LAB, as well as vitamins to a lesser extent. Besides commercial media, such as De Man Rogosa Sharpe (MRS), milk, and cheese whey, soybean whey supplemented with carbohydrate sources or nitrogen sources can also be used for the propagation of LAB, which seems to be a readily available, relatively low-cost and promising alternative medium for the incubation and fermentation of LAB. The components of the medium and carbohydrate source showed significant effects on the production of EPS, but the preferences for the maximum EPS yield and sugar composition were strain dependent [9]. Soybean whey wastewater (SWW) refers to the residual liquid produced as a result of the traditional production of soy protein isolate (SPI) [10]. A large amount of SWW is discharged, even about 20 times the yield of SPI. However, SWW is enriched in several valuable compounds such as oligosaccharides, protein, isoflavones and inorganic salts, which are necessary for the growth of microorganisms [11]. SWW is also regarded as a high-concentration industrial wastewater due to its high values of chemical and biological oxygen consumption [12]. It is favorable to the growth of microorganisms, making it prone to putrefaction and causing serious environmental pollution. Meanwhile, the heavy processing volume of soybean products in China has aggravated the burden of wastewater treatment by enterprises. In recent years, the value-added and full utilization of SWW in an environmentally friendly, economical and efficient manner, especially the biological treatment method, has received more attention. Extensive studies have employed SWW as the medium to produce active substances through fermentation [13,14,15]. However, little attention has been paid to studies on the conversion of waste into treasure by using SWW as a fermentation medium to produce functional metabolites.

In this study, optimized SWW-enriched 10% sucrose was used for the incubation and fermentation of *Lc. citreum* 1.2461. The primary structural characterization of purified EPS-1 isolated from *Lc. citreum* 1.2461 was determined by high-performance liquid chromatography (HPLC), flourier transform infrared (FT-IR) spectroscopy, one- (1D) and two- (2D) dimensional nuclear magnetic resonance (NMR) spectroscopy and scanning electron microscope (SEM). Additionally, the rheological properties of the water-soluble EPS-1 were further evaluated. We also investigated the in vitro antioxidant activities and potential protective effects on yeast resistance to oxidative stress of EPS-1.

## 2. Materials and Methods

### 2.1. Microorganism and Chemicals

*Lc. citreum* 1.2461 was obtained from China General Microbiological Culture Collection Center (CGMCC 1.2461). DEAE-cellulose-52 was purchased from Whatman Co., Ltd. (Maidstone, Kent, UK). Dialysis membrane (Mw cut-off 8000–14,000 Da) was gained from Solarbio Co., Ltd. (Beijing, China). 2,2′-azino-bis (3-ethylbenzothiazoline-6-sulfonic acid) (ABTS), Tri-2-pyridyl-s-triazine (TPTZ), vitamin C (Vc), 1,10-phenanthroline, arabinose, mannose, rhamnose, fructose, glucose and galactose were bought from Sigma (St. Louis, MO, USA). All other reagents used were analytical grade

### 2.2. Determination of Growth Curve

Cultivations of *Lc. citreum* 1.2461 for EPS were conducted at 37 °C with inoculum concentration of 4% (*v*/*v*) in optimized soybean whey-enriched 10% (*m*/*v*) sucrose. The initial colony count was approximately 6.5 lgCFU/mL. Samples of approximately 100 mL were collected at different time intervals (every 6 h). The pH values were evaluated in real time by a Schott pH Meter (model, SartoriusPB-10, Gottingen, Germany). The TA (°T, expressed as Thorner degree, °T × 0.009 = lactic acid%) of the different samples was performed based on AOAC [16]. The colony counts were determined with reference to national standard GB 4789.2-2016, and the optical value of samples was detected using spectrophotometer at an absorbance of 600 nm. EPS was isolated using the method as described in Section 2.3.1.

### 2.3. EPS Characterization

#### 2.3.1. Production of EPS

EPS was isolated based on the previous method [17]. Briefly, fermentation broth was collected and centrifuged at 12,000 rpm/min to remove cells and denatured proteins. Then, the supernatant was added with trichloroacetic acid (TCA) to the final concentration of 4% (*w*/*v*) and kept at 4 °C for 4–6 h followed by centrifugation to remove the protein. The EPS was then precipitated with triploid volume of 98% ethanol for 8–12 h. After centrifugation, the precipitate was suspended and dialyzed in distilled water for 48 h using dialysis bag (Mw cut-off: 8000–14,000 Da). The crude EPS was collected after lyophilization for 2 days. Furthermore, the freeze-dried crude EPS (10 mg/mL) was further purified by a DEAE-52 anion exchange column (2.6 × 30 cm) and eluted stepwise with deionized water, 0.1 and 0.3 M NaCl. The total sugar content was evaluated by phenol–sulfuric acid method. After dialysis and lyophilization, the obtained fraction with higher content (80.2%) was named EPS-1 for further study.

#### 2.3.2. Chemical Composition Analysis

Total sugar content was determined by the phenol–sulfuric acid method using D-glucose as standards [18]. Protein content was evaluated using the Bradford Protein Assay Kit (Sparkjade Biotech Co., Ltd., Shandong, China) using bovine serum albumin (BSA) as standard. Uronic acid content was measured by the carbazole–sulfuric acid method using glucuronic acid as standard [16].

#### 2.3.3. Monosaccharide Composition Analysis and Molecular Weight (Mw) Determination

The monosaccharide composition of EPS-1 was performed by HPLC with complete acid hydrolysis method [19]. In brief, 5 mg of the purified EPS-1 was hydrolyzed with 2 mL trifluoroacetic acid (TFA, 2M) at 120 °C for 2 h. Then, the sample added with a small amount of methanol (2 mL, repeat 3 times) was steamed to remove the residual TFA, affording the PMP derivative of EPS-1. Six monosaccharide standards (fructose, Fru; mannose, Man; rhamnose, Rha; glucose, Glc; galactose, Gal and arabinose, Ara) were treated the same way as above. Calculation of the molar ratio of the monosaccharide composition was accomplished according to the peak area. The monosaccharide composition and molar ratio of EPS-1 were determined by comparing the retention time and chromatographic peak area of standard monosaccharide derivatives and EPS-1 derivative. The homogeneity and Mw of EPS-1 was characterized by High performance size-exclusion chromatography (HPSEC). The linear curve was calibrated with standard T-series dextrans.

#### 2.3.4. Spectra Analysis

UV–Vis spectroscopy analysis of EPS-1 was conducted using a UV-1603 spectrophotometer (Shimadzu Co., Ltd., Kyoto, Japan) in the wavelength range of 190–500 nm; distilled water was used as the blank control. Dried EPS-1 samples (1.0 mg) were mixed with potassium bromide (KBr, 100.0 mg) and pressed into transparent flakes. FT-IR spectrum of EPS-1 was collected in the wavelength range of 4000–400 cm^−1^ on a FT-IR spectrophotometer (Bruker Co., Ettlingen, Germany).

#### 2.3.5. Congo Red Test

The conformational structure of the EPS-1 was supposed by helix-coil transition analysis. The EPS-1 sample was dissolved in 0.15 M NaOH solution containing 0.1 mM Congo red to a final concentration of 1 mg/mL. The mixture was scanned in the range of 190–800 nm after 3 h of reaction at 25 °C. The Congo red solution without sample was considered as the control group, and the maximum absorption wavelength of Congo red solution with EPS-1 sample was observed to determine a red shift.

#### 2.3.6. 1D- and 2D-NMR Spectra Analysis

The structure of EPS-1 was determined by ^1^H NMR, ^13^C NMR and 2D-NMR spectra on a AVANCE AV-500 spectrometer (Bruker Group, Fällanden, Switzerland) spectrometer at 500 MHz. The freeze-dried EPS-1 sample was dissolved in D_2_O and analyzed. The coupling constants (*J*) were given in hertz, and the chemical shifts (*δ*) were described in parts per million (ppm). The sequence of sugar residues was confirmed by the signals assigned by the 2D-NMR measurements.

#### 2.3.7. Scanning Electron Microscopy (SEM) Analysis

The morphology of EPS-1 was observed using a HITACHI S-3000N emission scanning electron microscope (Hitachi, Science Systems, Ltd., Hitachinaka, Japan). The dried sample was placed on an aluminum stub by using stick tape and then sputtered with gold (10 nm). Micrographs were recorded at 400×, 1000×, 2000×, 3000× magnification at an accelerating voltage from 400 V to 3.0 kV.

### 2.4. Rheological Properties of EPS-1

Rheological properties of EPS-1 from *Lc. citreum* 1.2461 were evaluated according to our previous method using a shear rate-controlled rheometer (Physica MCR 301, Anton-Paar, GmbH, Graz, Austria) with Model PP50 stainless steel parallel plates at 1 mm gap. Briefly, different concentrations of EPS-1 (2%, 4% and 6%, *w*/*v*) samples were prepared, and the EPS-1 samples were dissolved in different pH values aqueous solutions (3.0, 7.0 and 10.0). Meanwhile, Na^+^, K^+^, Ca^+^ and Mg^2+^ at a concentration of 8% (*w*/*v*) were mixed separately with EPS-1 sample (2%, mg/mL). The influence of sample concentration, pH and ions on apparent viscosity of EPS-1 were detected separately in the frequency range of 0.1–100 Hz at room temperature. Furthermore, the effects of temperature on the apparent viscosity of EPS-1 (2%, 4% and 6%, *w*/*v*) were evaluated in the scanning range from 20 to 80 °C (heating at the rate of 2 °C/min).

### 2.5. Antioxidant Properties of EPS-1

#### 2.5.1. ABTS Free Radical Scavenging Activity

The ABTS radicals scavenging activity of EPS-1 was measured with reference to Lobo et al. [20]. The sample solution with different concentrations (1–12 mg/mL) was added to a tube containing 0.4 mL of ATBS radical solution (7 mM) and 0.8 mL of K_2_S_2_O_8_ solution (2.45 mM). Then, the mixture was shaken thoroughly and incubated at 25 °C, and the absorbance was measured at 734 nm after 20 min.VC was used as the positive control.

#### 2.5.2. Ferric Ion Reducing Activity

The ferric ion reducing ability of EPS-1 was determined using the approach mentioned by Si et al. [21]. The FRAP reagent was performed and incubated at 37 °C. The EPS-1 (1–12 mg/mL) was then mixed with FRAP reagent (1:9 volume ratio) and incubated for 10 min at 37 °C; the absorbance was measured at 593 nm. The results were expressed in terms of Fe^2+^ concentration (μg/mL).

#### 2.5.3. Metal Ion Chelating Activity

The metal ion chelating activity of EPS-1 was conducted according to Liu et al. with slight modifications [22]. Ethylenediaminetetra acetic acid disodium salt (EDTA-Na) was used as the positive control.

### 2.6. Effect of EPS-1 on Yeast Resistance to Oxidative Stress

The effect of EPS-1 on yeast resistance to oxidative stress was performed by the previous method with slight modification [23,24]. Yeast cells were incubated in yeast extract peptone dextrose (YPD) medium to the exponential phase (OD_600_ = 0.5–0.6) and centrifuged at 5000 rpm for 5 min. Then, the collected cells were suspended in 100 mM PBS (pH = 7.4). EPS-1 solutions with different concentrations (1–12 mg/mL) were added and cultured at 28 °C with shaking for 18 h, and subsequently exposed to 250 mM hydrogen peroxide (H_2_O_2_) for 1 h. Cell viability was evaluated by the microplate reader at intervals of 6 h. In addition, the supernatant of samples was detected using superoxide dismutase (SOD) activity assay kit, catalase (CAT) assay kit and reduced glutathione (GSH) assay kit following the manufacturer’s protocols. Cells untreated with H_2_O_2_ were used as control group1 (CK1), cells untreated with H_2_O_2_ but added with EPS-1 were used as control group 2 (CK2), while the model group (model control) represented cells treated with H_2_O_2_ but not protected by EPS-1. All the samples were examined microscopically; cells in CK1 represented normal control.

### 2.7. Statistical Analysis

All experimental results were presented as mean ± standard deviations (SD). The obtained data were analyzed by one-way analysis of variance (ANOVA) and Fisher’s least significant difference (LSD) tests using SPSS 22.0 statistical software. A probability of *p* < 0.05 was considered as statistically significance.

## 3. Results and Discussion

### 3.1. Time Course of Bacterial Growth and EPS Production by Lc. citreum 1.2461

*Lc. citreum* 1.2461 was inoculated in optimized soybean whey-enriched 10% sucrose and cultured at a constant temperature. The changes in viable bacteria number, optical value, acid production and EPS production are shown in Figure 1. The number of viable bacteria reached the highest level at 18 h, about 7.78 lgCFU/mL. After that, the total number of bacteria gradually decreased with the change of fermentation time. The pH value of the soybean whey fermentation medium was always decreasing during the whole fermentation process and reached a stable value after 12 h. The pH value at the end point of fermentation was 4.37. Furthermore, the titration acidity value changed rapidly within the first 12 h, indicating that the strain was in the logarithmic phase of growth with strong acid production capacity. Additionally, the production of total EPS increased rapidly during the first 24 h and reached a maximum yield of 8.32 g/L at 24 h. These results suggest that EPS production was seemingly related to bacterial growth and fermentation time.

### 3.2. EPS-1 Characterization

#### 3.2.1. Isolation and Purification of EPS-1

The crude EPS extracted from *Lc. citreum* 1.2461 was fractionated through DEAE-Cellulose 52 anion-exchange chromatography, affording 80.2% of EPS-1 (Figure 2A). The sugar contents in crude EPS and EPS-1 were 87.68% and 94.76%, respectively. Meanwhile, a certain degree of uronic acid (3.42%) and protein (1.54%) was detected in crude EPS, suggesting the presence of protein impurities in the crude EPS (data not shown). However, contents of protein and uronic acid were not detected in EPS-1 after purification.

#### 3.2.2. Monosaccharide Composition and Mw of the EPS-1

The HPLC chromatograms of each standard monosaccharide sample derived by PMP measured under detection conditions are shown in Figure 2B. Compared with the standard monosaccharide, the EPS-1 isolated from *Lc. citreum* 1.2461 was basically composed of glucose residues. The purity and Mw of EPS-1 were further determined by HPSEC. It could be seen that EPS-1 from *Lc. citreum* 1.2461 had single symmetrical peaks in the chromatogram (Figure 2C). In addition, the average Mw was calculated to be 4.71 × 10^6^ Da according to a calibration curve (Log *Mw* = −2.4743*T*_R_ + 25.181, *R*^2^ = 0.9994, where *Mw* indicates the molecular weight, *T*_R_ indicates retention time) gained from standard T-series dextrans with different Mw, which was comparable to EPS found in other *Leuconostoc* strains (range of 10^6^–10^7^) [25].

#### 3.2.3. UV–Vis, FT-IR Analysis and Congo Red Test

The ultraviolet full-wavelength scanning spectrogram of the purified EPS-1 is illustrated in Figure 3A. It does not show absorption at 260 or 280 nm from the UV spectra, demonstrating the absence of nucleic acid or protein in EPS-1. The result was also consistent with the analysis of the conventional components of the EPS-1 mentioned above. The purified EPS-1 showed typical polysaccharide absorption peaks in the FT-IR spectrogram (Figure 3B). The strong and wide absorption peak of EPS-1 near 3368 cm^−1^ was attributed to the stretching vibrations of the O–H bond. The absorption bonds at 2931 cm^−1^ can be ascribed to the C–H stretching vibration, and the peak at 1649 cm^−1^ was due to the stretch vibration of C=O bond. [26]. Moreover, the peaks at 918 and 849 cm^−1^ suggest the existence of α-1,3 linkages, and the sharp absorption peak at approximately 1020 cm^−1^ is a characteristic of the α-1,6 linkage, which could be further verified by NMR analysis [27,28]. The λ_max_ of the Congo red-EPS-1 mixture of the 0.15 M NaOH solution was measured by UV–Vis, as shown in Figure 3C. It can be observed that the maximum absorption wavelength of EPS-1 was not reduced significantly after the reaction with the Congo red solution, suggesting that the EPS-1 fraction had a non-three helical structure.

#### 3.2.4. NMR Spectroscopy Analysis

The EPS-1 from *Lc. citreum* 1.2461 was further investigated using 1D- and 2D-NMR spectroscopy techniques, including ^1^H, ^13^C and ^1^H-^13^C HSQC NMR experiments. In the ^1^H NMR spectrum (Figure 4A), the intensive peak at *δ*4.70 ppm was due to HDO, and two clear signals from anomeric protons (H-1) were observed at *δ*5.28 and *δ*4.93 ppm (named **A** and **B**), which were probably attributed to α-1,3 and α-1,6 linked D-glucopyranose units, respectively [29,30]. The ratios of the α-(1→3) linkage and α-(1→6) linkage were approximately 1:4, obtained from the integration analysis. Compared with previous studies, the EPS from *Lc. citreum* KM01 was confirmed to be α-1,6 linked dextran with a high-intensity anomeric signal at *δ*5.27 ppm [25]. While three signals at *δ*4.90, *δ*5.14 and *δ*5.24 ppm were found in EPS from *Lc. citreum* B-2 was assigned to α-(1→6) glucosyl residues, α-(1→2) glucosyl residues and α-(1→3) glucosyl residues, respectively [7]. In ^13^C NMR spectrum (Figure 4B), two major anomeric carbon resonances at *δ*102.69 and *δ*100.36 ppm corresponded to H-1 anomeric protons at *δ*5.28 and *δ*4.93 ppm, respectively. The two peaks at *δ*81.16 and *δ*68.26 ppm were clear signals of the downfield shift of the α-1,3 and α-1,6 linked glucosyl residues. Additionally, the signals at chemical shifts *δ*76.05, *δ*74.07, *δ*72.85 and *δ*72.20 were assigned to C-3, C-2, C-5 and C-4 glucosyl residues, respectively [31]. Furthermore, the complete assignments of ^1^H and ^13^C signals for EPS-1 were identified based on the 2D ^1^H-^13^C HSQC (Figure 4C). As shown in Table 1, residue **A** had a C-3 signal at *δ*81.16 ppm and an attached H-3 signal at *δ*3.65 ppm, which was downfield of the same nucleus in residue **B** (*δ*76.05 and *δ*3.68 ppm). Similarly, the C-6 signal of residue **B** at *δ*68.26 ppm cross-linked with the H-6 signal at *δ*3.92 ppm, showing a lower field than residue **A** (*δ*63.19 and *δ*3.73 ppm). Taken together, the EPS-1 from *Lc. citreum* 1.2461 was composed of →3)-α-D-Glcp-(1→ and →6)-α-D-Glcp-(1→, and the predicted repeating unit of EPS-1 was determined, as shown in Figure 4D.

#### 3.2.5. SEM Analysis

The scanning electron microscope is considered as a powerful tool to elucidate the surface morphology of macromolecular polymers and inference of their physical properties. The microstructure and surface morphology micrographs of EPS-1 at different magnifications (400×, 1000×, 2000× and 3000×) is illustrated in Figure 5. As observed, the EPS-1 particles were seemingly sheet-like with irregular structure. However, the surface morphology of EPS-1 was smooth and relatively compact, which could be related to the strong intermolecular interaction of polysaccharide, making it suitable for application as thickening, stabilizing and emulsifying agents during food storage. The surface morphology of the EPS-1 was similar to that of EPS produced by *Lc. citreum* B-2, which revealed the compact and glittering surface [7].

### 3.3. Rheological Measurement

In Figure 6, the apparent viscosity of EPS-1 from *Lc. citreum* 1.2461 was influenced by shear rate, confirming that the EPS-1 solution had the characteristics of shear thinning and was a typical pseudoplastic non-Newtonian fluid. Singthong et al. believed that the ordered structure could be formed between polysaccharide molecules through chain–chain entanglement [32], and the shear force could destroy the ordered structure to reduce the viscosity. As can be seen from Figure 6A, the apparent viscosity of EPS-1 was improved with the increasing concentrations of the sample from 2% (*m*/*v*) to 6% (*m*/*v*). However, the trend of rheological curve of the EPS-1 solution did not change significantly with the adjustment of pH values or the addition of metal ions (Figure 6B-C), indicating that EPS-1 is relatively stable in acidic and alkaline solutions. At the same time, it was speculated that the EPS-1 could not form gels from interacting with metal ions, demonstrating the good stability of salt tolerance and promising application in high-salt foods for EPS-1. Furthermore, the EPS-1 showed a gradual reduction in the apparent viscosity in response to the increasing temperature in Fig 6D. It was considered that the enhanced intensity of the molecular motion and diffusion during the heating process led to the reduction of intermolecular force and solution viscosity. However, the apparent viscosity was increased at first and later declined for the EPS-1 solution at 4% and 6% (*w*/*v*) concentrations, which could be related to the slow solubility of the high concentration of EPS-1 at room temperature. Viscosity, concentration, temperature, shear rate, coexisting ions and other factors will affect the fluid characteristics of polysaccharide colloids in food processing processes [33]. Therefore, the results provided a theoretical support for the potential application of EPS-1 in the field of food.

### 3.4. Antioxidant Activities of EPS-1

The antioxidant properties of EPS-1 from *Lc. citreum* 1.2461 were investigated in the concentration ranges from 1 to 12 mg/mL, including ABTS radical scavenging ability, chelating activity on metal ions, and ferric ion reducing antioxidant ability. As shown in Figure 7A, the ABTS radical scavenging ability of EPS-1 was improved with the increase in concentration, and Vc showed relatively excellent scavenging ability on the ATBS radical when compared with EPS-1. The scavenging effect of EPS-1 increased from 2.81% to 33.34% with the increasing concentrations. In addition, the antioxidant capacity of EPS-1 on ferric ion reducing was relatively weak compared with that of Vc, and the increase in concentration was not accompanied by the enhancement of antioxidant activity (Figure 7B). Moreover, the chelating effect of EPS-1 on metal ions was in a concentration-dependent way. At the concentration of 12 mg/mL, the chelating activity of EPS-1 on metal ions was at 42.4% (Figure 7C). Notably, the EPS-2 from *L. helveticus* MB2-1 and its derived polysaccharide exhibited chelating activity of more than 60% at 4 mg/mL [34]. The research shows that EPS with the antioxidant activities may influence anticancer activity, and further studies should focus on the antioxidant mechanisms of EPS through in vitro and in vivo models [35]. In summary, the antioxidant activities of EPS-1 gradually strengthened with the increasing concentrations of the sample, showing a significant dose–effect relationship.

### 3.5. Effect of EPS on Yeast Resistance to Oxidative Stress

Oxidative stress plays a vital role in the pathogenesis of many diseases. Generally, antioxidant enzymes can slow the oxidative damage during antioxidant stress by means of stabilizing the excess electrons of free electrons [36,37]. In this study, the oxidative stress reaction of yeast cells was detected under the condition of H_2_O_2_ (250 mM). In Figure 8A, cells treated with H_2_O_2_ alone (model control) exhibited acute damage as confirmed by a significant decline in the average optical density (OD_600_). The protective effect of EPS-1 on yeast cells strengthened with increasing concentrations, especially at a high concentration of 12 mg/mL, and the survival rate of yeast cells was up to 54% with the protection of EPS-1. It is evident that yeast cells of the normal control were in good growth condition and exhibited a spherical shape as shown in Figure 8B. In contrast, the yeast cells in the model group (model control) were almost completely damaged with few intact cells visible microscopically. With the increasing concentration of EPS-1, the amount of damaged cells decreased gradually, and most yeast cells tended to return to normal with concentrations of EPS-1 at 12 mg/mL. As illustrated in Figure 9, the activities of SOD, CAT and GSH-Px activities in H_2_O_2_-injured cells (model control) were significantly declined compared with the normal control group. After incubation by adding different concentrations of EPS-1 solutions, the enzyme activity of each group showed an increasing trend. Moreover, the promotion effect of EPS-1 on CAT and GSH-Px activities showed a dose-dependent relationship. At concentrations of 12 mg/mL, the protection ability of EPS-1 was superlative, and the amounts of CAT and GSH-Px reached 8.88 and 7.90 U/mL, respectively. The level of SOD in cells with EPS-1 was markedly promoted in the level of SOD at 4 mg/mL and decreased with the increasing concentrations. Taken together, in vitro models of antioxidant activity have proven that EPS-1 from *Lc. citreum* 1.2461 could slow down the oxidation process by regulating antioxidant enzymes, such as SOD, CAT and GSH-Px.

## 4. Conclusions

In the present work, the optimized soybean whey-enriched 10% (*m*/*v*) sucrose was utilized for the incubation and fermentation of *Lc. citreum* 1.2461. The purified EPS fraction (EPS-1) was then isolated and characterized by UV, FT-IR, HPLC and NMR analyses. The structural analysis suggested that EPS-1 was mainly composed of α-1,3 and α-1,6 linked D-glucopyranose units with *Mw* of 4.71 × 10^6^ Da. The unique sheet-like compact microstructure of EPS-1 and rheological properties could be beneficial to its potential application in food and other diverse industries. Moreover, EPS-1 was demonstrated to have certain antioxidant abilities. To better understand the structure–bioactivity correlation in the EPS-1 from *Lc. citreum* 1.2461, further studies are necessary.

## Figures and Tables

**Figure 1 foods-11-02283-f001:**
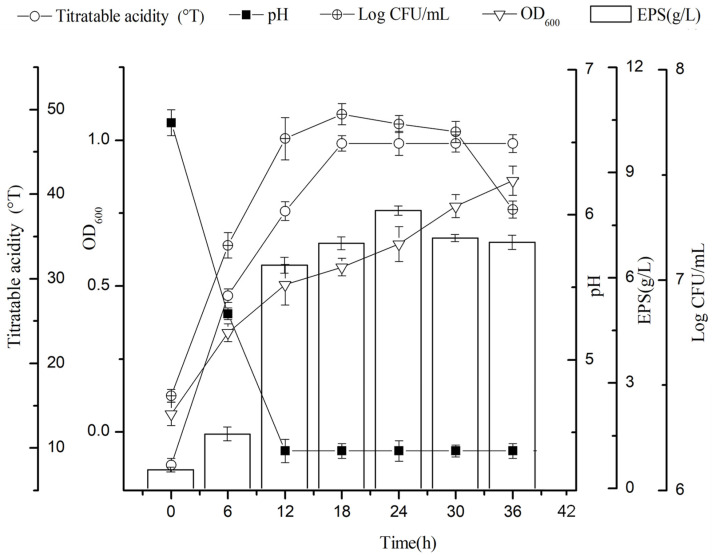
Growth curve of bacterial growth and EPS production by *Lc. citreum* 1.2461 at 37 °C from 0 to 36 h.

**Figure 2 foods-11-02283-f002:**
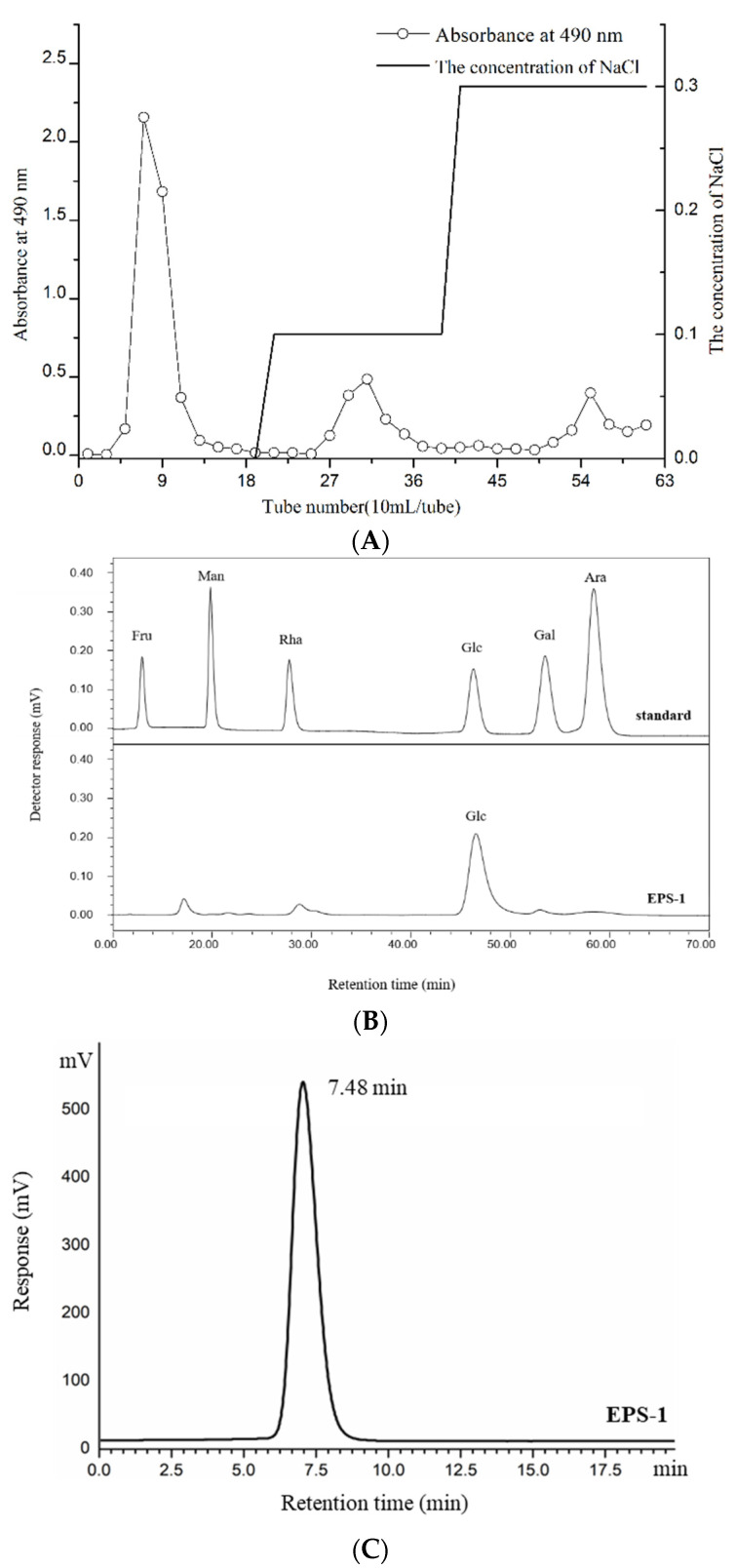
DEAE-cellulose-52 anion exchange chromatogram (**A**), monosaccharide composition chromatogram (**B**) and HPSEC chromatogram of relative Mw distribution (**C**) of EPS-1 from *Lc. citreum* 1.2461.

**Figure 3 foods-11-02283-f003:**
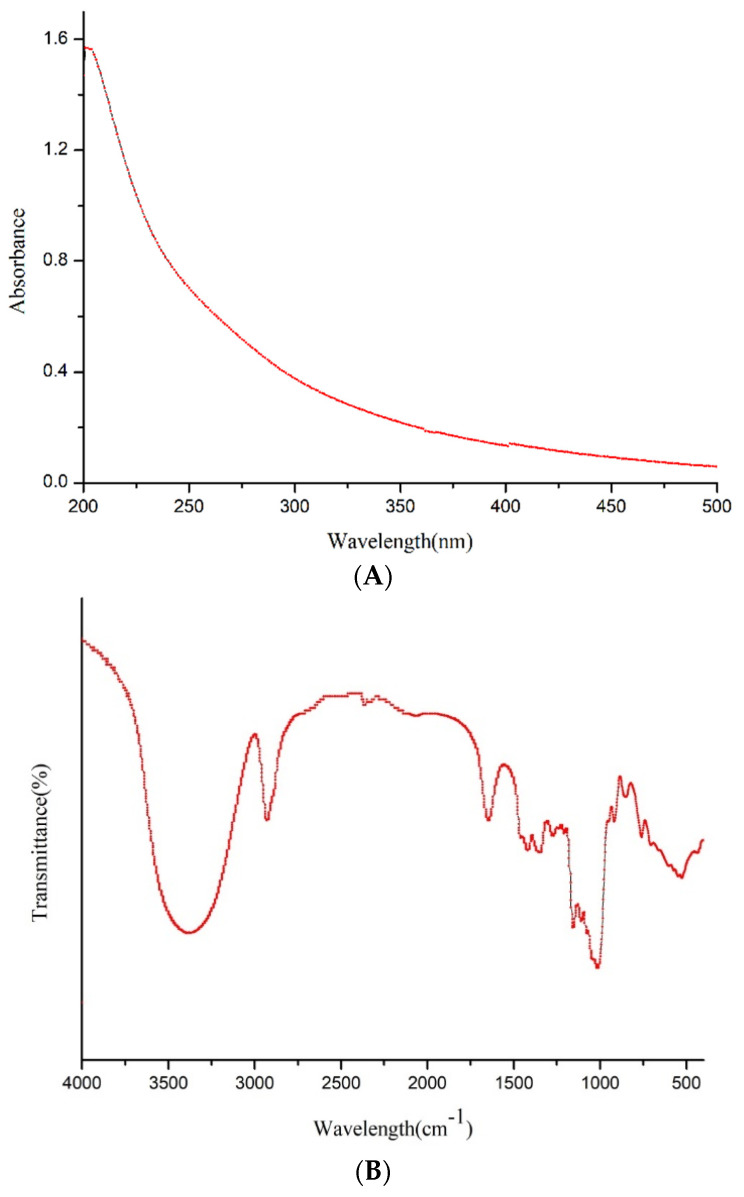
UV-Vis spectrogram (**A**), FT-IR spectrogram (**B**) and Congo red test analysis (**C**) of EPS-1 from *Lc. citreum* 1.2461.

**Figure 4 foods-11-02283-f004:**
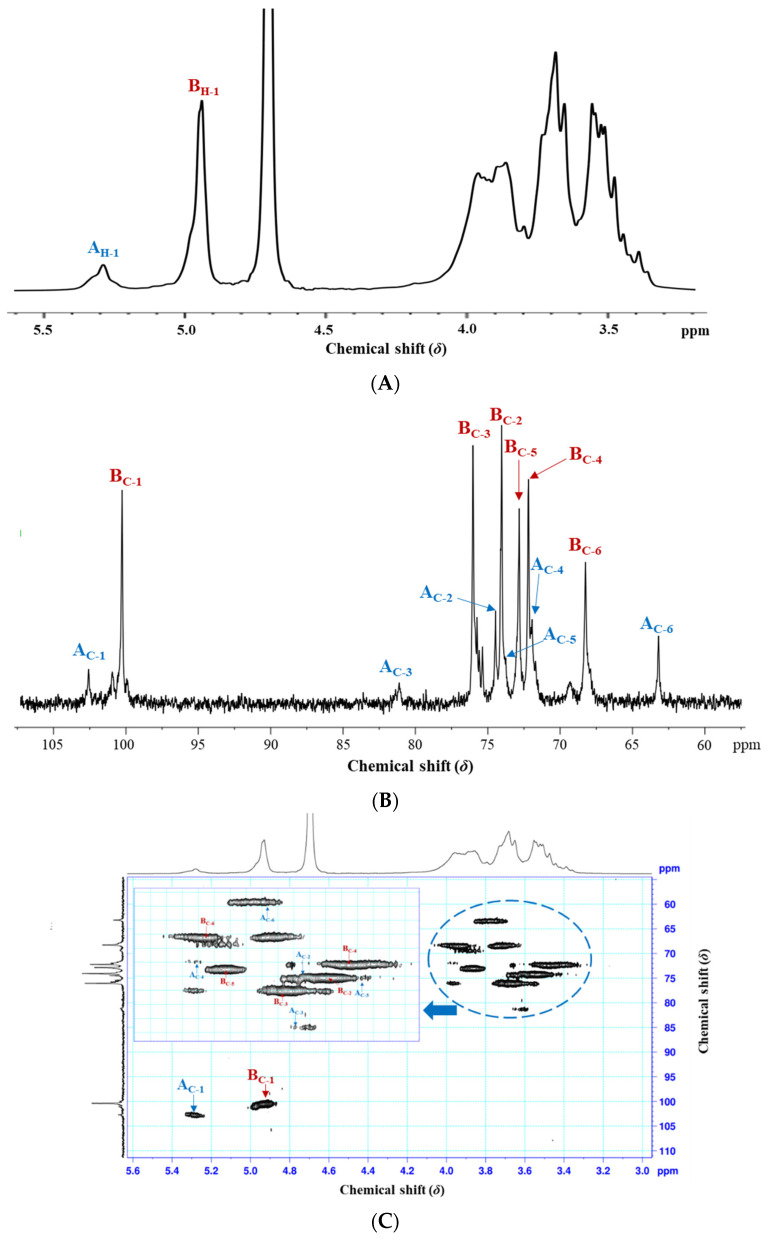
The ^1^H NMR (**A**), ^13^C NMR (**B**) and 2D ^1^H-^13^C HSQC (**C**) spectrograms and proposed structure (**D**) of EPS-1 from *Lc. citreum* 1.2461.

**Figure 5 foods-11-02283-f005:**
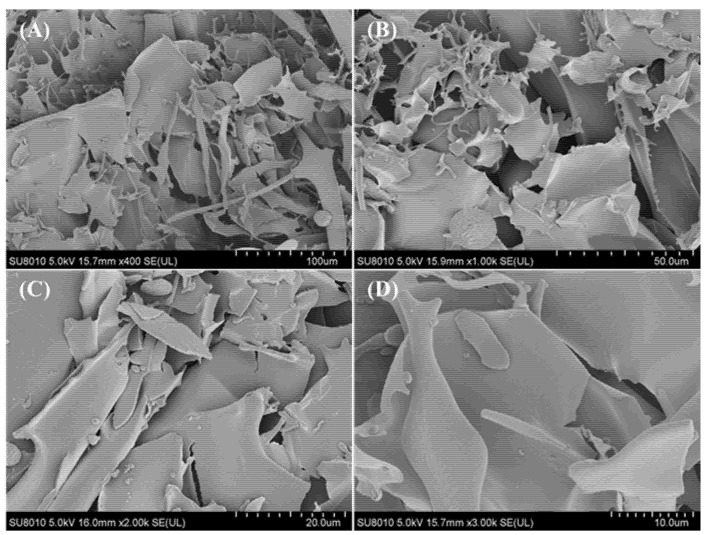
Scanning electron micrographs of EPS-1 from *Lc. citreum* 1.2461. 400× (**A**), 1000× (**B**), 2000× (**C**) and 3000× (**D**).

**Figure 6 foods-11-02283-f006:**
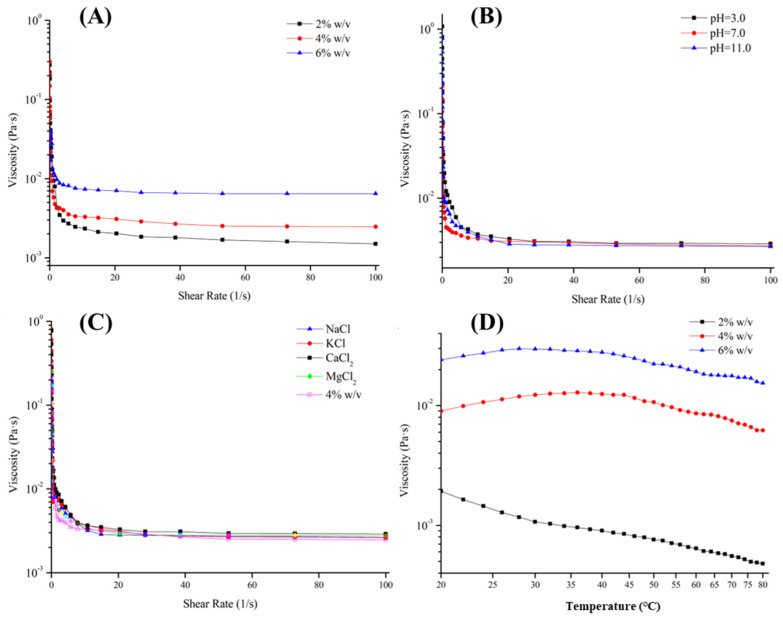
Effects of polysaccharide concentration (**A**), pH value (**B**), metal ion (**C**) and temperature (**D**) on the apparent viscosity of EPS-1 from *Lc. citreum* 1.2461.

**Figure 7 foods-11-02283-f007:**
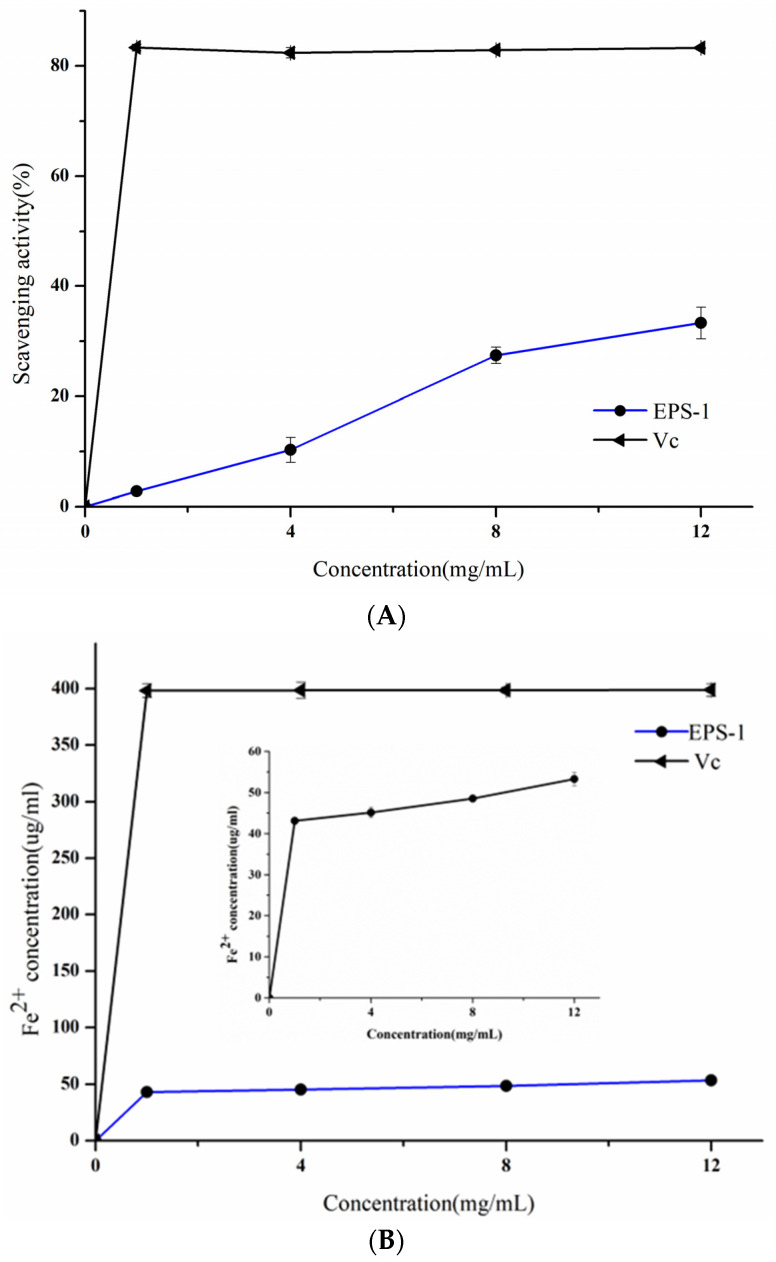
Scavenging activities on ABTS radical (**A**), reducing antioxidant ability on ferric ion (**B**) and metal ion chelating activity (**C**) of EPS-1 from *Lc. citreum* 1.2461.

**Figure 8 foods-11-02283-f008:**
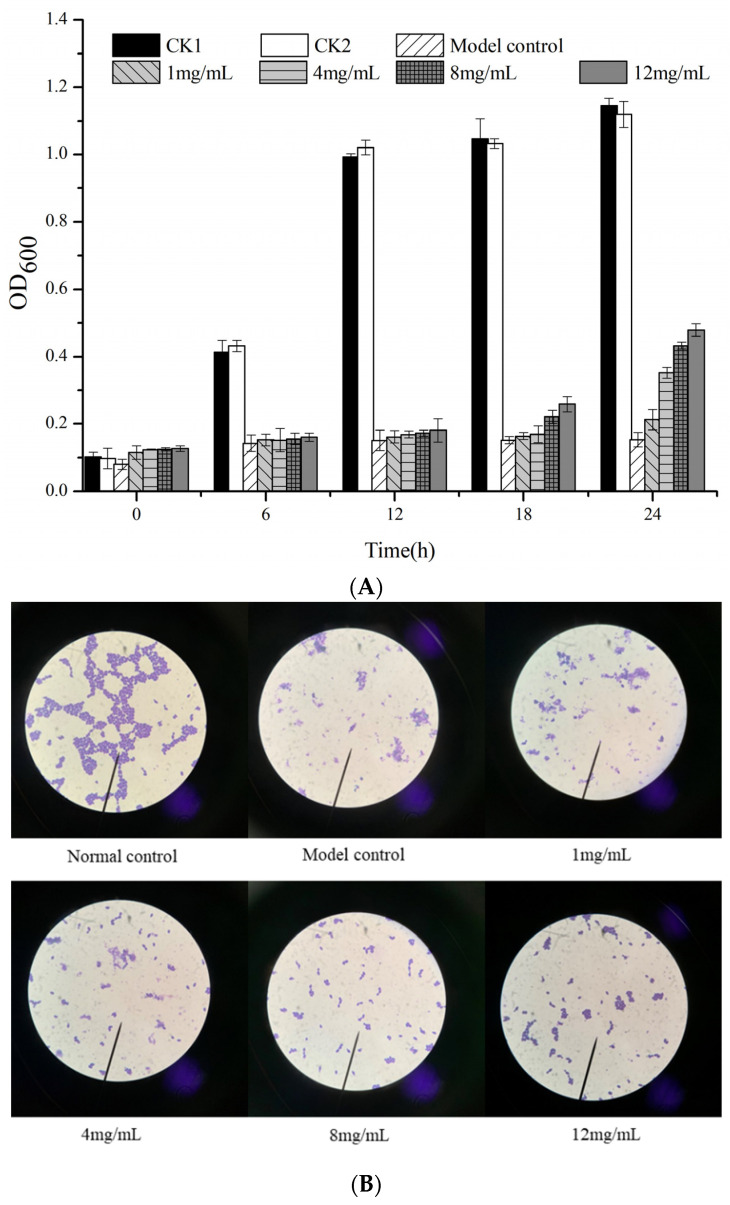
Effect of EPS-1 from *Lc. citreum* 1.2461 on yeast protection to oxidative damage in vitro (**A**). Different experimental groups were observed microscopically (**B**).

**Figure 9 foods-11-02283-f009:**
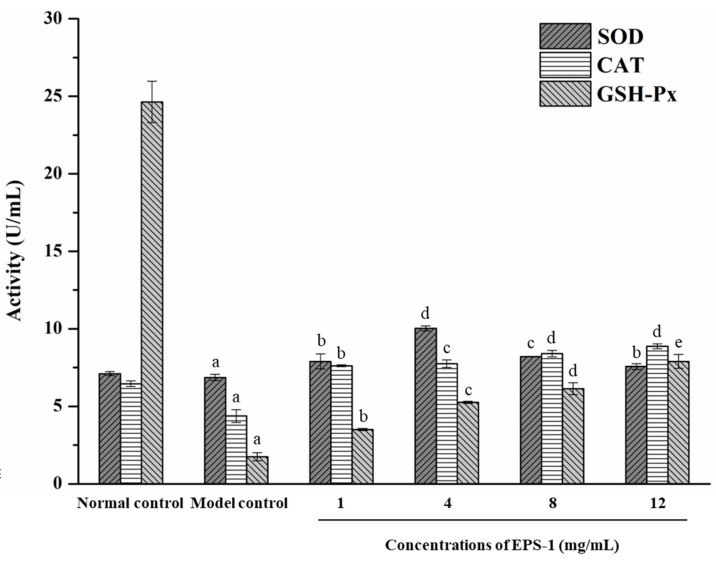
Effects of EPS-1 from *Lc. citreum* 1.2461 on the SOD, CAT and GSH-Px activities of yeast protection of oxidative damage. Data shown represent the mean ± SD of three replicates. Different lower case letters (a–d) in superscript denote significant difference (*p* < 0.05).

**Table 1 foods-11-02283-t001:** Chemical shifts (ppm) of ^1^H and ^13^C signals for EPS-1 from *Lc. citreum* 1.2461.

Glycosyl Residues	H-1/C-1	H-2/C-2	H-3/C-3	H-4/C-4	H-5/C-5	H-6/C-6
**A**	5.28	3.62	3.65	3.95	3.44	3.73
→3)-α-D-Glcp (1→	102.69	74.49	81.16	71.96	73.79	63.19
**B**	4.93	3.54	3.68	3.47	3.86	3.92
→6)-α-D-Glcp (1→	100.36	74.07	76.05	72.20	72.85	68.26

## Data Availability

The data used to support the findings of this study can be made available by the corresponding author upon request.

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
