# Peer review of "Structural Characterization, Rheological Properties and Protection of Oxidative Damage of an Exopolysaccharide from Leuconostoc citreum 1.2461 Fermented in Soybean Whey"

_foods, 2022, doi:10.3390/foods11152283_

Round 1

Reviewer 1 Report

The study "Structural characterization, rheological properties and protection of oxidative damage of an exopolysaccharide from Leucoostoc citreum 1.2461 fermented in soybean whey" represent interesting research, however, it is necessary to make some corrections:

ABSTRACT: The abstract lacks structure.  There is no background, objective, justification, or conclusion of the study. This is limited only to the explanation of the experiments and the results. It is necessary to restructure. 

Line 34: The cited article does not mention exopolysaccharides of microbial origin. Redefine the phrase or change the reference

Line 37-38: Change the expression "...EPS was linked..." by "EPS has been linked..."

Line 60: change by "... SWW is discharged after "

Line 74: It probably is better if you use the word "enriched" instead  of "with the addition"

Redefine L. citreum in all text by Lc. citreum, this is the correct form for abbreviation 

Line 92: Why did the authors define the analysis as a growth kinetics curve, while they didn't calculate kinetic constants? It probably is better if they eliminate the word "kinetics". But, if they decide to keep it it is necessary to show the comparison of each kinetic parameter in all curves determined for all carbon sources analyzed. 

Line 93: The initial volume of the inoculum was not defined and it is not known if the experiment is from a single fermentation unit or if they are independent units. That is, was the sampling carried out from a single fermentation unit?

Line 94: The authors must define the concentration of the inoculum based on the CFU per mL

Line 94-95: How many intervals of time were chosen by sampling? 

Line 106: "And" should be eliminated  to start the phrase with "Then"

Line 110-111: What were the lyophilization conditions? What was the brand or type of lyophilizer mechanism?

Line 114: The phenol-sulfuric acid method needs a reference

Line 213-214: In all text, try to use the word enriched instead whit an addition of 10%...

Line 216: the authors mention that the highest concentration of microorganisms was found at 18 hours, however, they did not consider the statistics of the graph. There appears to be no significant difference throughout the stationary phase, which begins at 12 hours. Its maximum concentration is reached at 12 hours and not at 18.

Fig 1: Since most of the changes occur before 36 hours, it is unnecessary to present data after this time, especially if there is a sampling gap at 42 hours. It is recommended to finish the graph up to 36 hours and explain in methodology

Fig 2: According to the authors, the chromatograms show that it is composed of glucose (fig 2C). However, Fig. 2B shows peaks that correspond, albeit at low concentrations, to other monosaccharides. In fact, through HPSEC analysis the molecular weight does not reflect the homogeneity of the structure. Please clarify and change the discussion.

Lines 255-256: the correct form to describe contamination absence is "... it is not shown absorption in the range 260 to 280, demonstrated protein absence in the  structure of EPS-1"

Line 261: the 1649 cm-1 absorption is not associated with water bending vibration. That absorption is related to C=O and carboxyl groups.  Please verify this part of the discussion.  The reference cited by the authors reflects what I have suggested

Section 3.4: the authors neglected to compare these results with literature data. If they do not exist, they must express them in the discussion. However, different studies have revealed the importance of antioxidant activity in exopolysaccharides of microbial origin. Please review: https://doi.org/10.1016/j.ijbiomac.2019.09.192 and https://doi.org/10.1016/j.ijbiomac.2021.10.047

Conclusion: I consider that all the characterization work is overshadowed by the last part of the article. I do not think the study protection against oxidative stress damage to yeast cells must be necessary. It appears to be an independent study that might as well be reported that way (independent). This also does not have a great impact on the conclusion that has been drawn up. I think that the conclusion should be rewritten because not all the results have been highlighted to describe the novelty of the research.

Mandatory Specification: Microorganism identification and NCBI entry number must be disclosed. How is it that the authors ensured the purity of the strain?

Author Response

Response to Reviewer 1 Comments

The study "Structural characterization, rheological properties and protection of oxidative damage of an exopolysaccharide from Leucoostoc citreum 1.2461 fermented in soybean whey" represent interesting research, however, it is necessary to make some corrections:

Point 1: ABSTRACT: The abstract lacks structure. There is no background, objective, justification, or conclusion of the study. This is limited only to the explanation of the experiments and the results. It is necessary to restructure.

Response: Thank you for your valuable suggestion, the abstract has been restructured in revised manuscript.

Point 2: Line 34: The cited article does not mention exopolysaccharides of microbial origin. Redefine the phrase or change the reference.

Response: Thank you for your nice caution, we have changed the reference in revised manuscript (Line 37).

Point 3: Line 37-38: Change the expression "...EPS was linked..." by "EPS has been linked..."

Response: According to your suggestion, “The bioactivity of EPS was linked to their specific structure” has been changed to “The bioactivity of EPS has been linked to their specific structure”. in revised manuscript (Line 40-41).

Point 4: Line 60: change by "... SWW is discharged after "

Response: Thank you. We have corrected the mistake according to your advice in revised manuscript (Line 62-63).

Point 5: Line 74: It probably is better if you use the word "enriched" instead  of "with the addition"

Response: Thank you for your suggestion, “the optimized SWW with the addition of 10% sucrose” has been changed to “the optimized SWW enriched with 10% sucrose” in revised manuscript (Line 77-78).

Point 6: Redefine L. citreum in all text by Lc. citreum, this is the correct form for abbreviation 

Response: Thank you. We have corrected the mistake in the full text according to your advice in revised manuscript.

Point 7: Line 92: Why did the authors define the analysis as a growth kinetics curve, while they didn't calculate kinetic constants? It probably is better if they eliminate the word "kinetics". But, if they decide to keep it it is necessary to show the comparison of each kinetic parameter in all curves determined for all carbon sources analyzed.

Response: Thank you for your valuable suggestion, we have eliminated the word “kinetics” to avoid ambiguity in revised manuscript (Line 95).

Point 8: Line 93: The initial volume of the inoculum was not defined and it is not known if the experiment is from a single fermentation unit or if they are independent units. That is, was the sampling carried out from a single fermentation unit?

Response: Thank you. The initial volume of the inoculum was 100 mL per triangular conical flask, and the experiment was divided into 10 independent units. After unified inoculation, they were put into the 37 incubator, and the sample was taken out every 6 hours for testing.

Point 9: Line 94: The authors must define the concentration of the inoculum based on the CFU per mL

Response: Thank you for your advice. The inoculum concentration was 4% (v/v) and the final concentration of the inoculum was approximately 6.5 lgCFU/mL in revised manuscript (Line 97-99).

Point 10: Line 94-95: How many intervals of time were chosen by sampling? 

Response: Thank you. The sample was taken out every 6 hours for testing.

Point 11: Line 106: "And" should be eliminated  to start the phrase with "Then"

Response: Thank you for your nice caution, we have corrected the mistake in revised manuscript (Line 110).

Point 12: Line 110-111: What were the lyophilization conditions? What was the brand or type of lyophilizer mechanism?

Response: Thank you. The crude EPS solution was frozen at −20 °C for 24 h and then was put into the freeze dryer chamber (Heto Power Dry LL 3000, Thermo Electron Corporation, Erie, PA, USA) at the pressure of 20 Pa. During the drying process, the temperature of heat shelf was set as 40 °C and the cold trap was set at lower than −50 °C.

Point 13: Line 114: The phenol-sulfuric acid method needs a reference

Response: Thank you for your suggestion, we have added a reference to support the phenol-sulfuric acid method in revised manuscript (Line 122).

Point 14: Line 213-214: In all text, try to use the word enriched instead whit an addition of 10%...

Response: Thank you. We have replaced the “with the addition of” with “enriched with” according to your advice in revised manuscript.

Point 15: Line 216: the authors mention that the highest concentration of microorganisms was found at 18 hours, however, they did not consider the statistics of the graph. There appears to be no significant difference throughout the stationary phase, which begins at 12 hours. Its maximum concentration is reached at 12 hours and not at 18.

Response: Thank you. We agree with the reviewer’s opinions. In fact, Lc. citreum 1.2461 could reach the stationary phase at 10 h according to the results of nonlinear fitting (data not shown). However, we just want to describe the time at which the maximum colony counts is reached based on the average number.

Point 16: Fig 1: Since most of the changes occur before 36 hours, it is unnecessary to present data after this time, especially if there is a sampling gap at 42 hours. It is recommended to finish the graph up to 36 hours and explain in methodology

Response: Thank you for your valuable suggestion. Fig 1 was modified in revised manuscript.

Point 17: Fig 2: According to the authors, the chromatograms show that it is composed of glucose (fig 2C). However, Fig. 2B shows peaks that correspond, albeit at low concentrations, to other monosaccharides. In fact, through HPSEC analysis the molecular weight does not reflect the homogeneity of the structure. Please clarify and change the discussion.

Response: Thank you. The EPS-1 was basically composed of glucose residues based on the Fig. 2B rather than Fig 2C. In addition, the peaks area assigned to other monosaccharides is less than one twentieth that of glucose. And the composition of EPS-1 was further determined by 1D- and 2D-NMR spectrum, which was composed of →3)-α-D-Glcp-(1→ and →6)-α-D-Glcp-(1→. Meanwhile, the controversial conclusion “indicating that EPS-1 was homogeneous polysaccharide.” were deleted in revised manuscript.

Point 18: Lines 255-256: the correct form to describe contamination absence is "... it is not shown absorption in the range 260 to 280, demonstrated protein absence in the structure of EPS-1"

Response: Thank you for your nice caution, we have corrected the mistake in revised manuscript (Line 260-261).

Point 19: Line 261: the 1649 cm-1 absorption is not associated with water bending vibration. That absorption is related to C=O and carboxyl groups. Please verify this part of the discussion.  The reference cited by the authors reflects what I have suggested

Response: Thank you. Taking the reviewer’s concern into account, we modified it as “The absorption bonds at 2931 cm-1 can be ascribed to the CH stretching vibration and the peak at 1649 cm-1 was due to the stretch vibration of CO bond.” in revised manuscript (Line 265-267).

Point 20: Section 3.4: the authors neglected to compare these results with literature data. If they do not exist, they must express them in the discussion. However, different studies have revealed the importance of antioxidant activity in exopolysaccharides of microbial origin. Please review: https://doi.org/10.1016/j.ijbiomac.2019.09.192 and https://doi.org/10.1016/j.ijbiomac.2021.10.047

Response: Thank you for your suggestion. We have added some comparison and discussion in revised manuscript (Line 357-361).

Point 21: Conclusion: I consider that all the characterization work is overshadowed by the last part of the article. I do not think the study protection against oxidative stress damage to yeast cells must be necessary. It appears to be an independent study that might as well be reported that way (independent). This also does not have a great impact on the conclusion that has been drawn up. I think that the conclusion should be rewritten because not all the results have been highlighted to describe the novelty of the research.

Response: In view of your valuable suggestion, we revised the content of conclusion.

Point 22: Mandatory Specification: Microorganism identification and NCBI entry number must be disclosed. How is it that the authors ensured the purity of the strain?

Response: Thank you. Lc. citreum 1.2461 was obtained from China General Microbiological Culture Col-lection Center. Preservation number of this strain was CGMCC 1.2461 =ATCC 49370 =CCUG 30060 =CIP 103315 =DSM 5577 =KCTC 3526 =LMG 9849. The NCBI reference sequence: NR_041727.1. We added the CGMCC preservation number in revised manuscript (Line 89).

Reviewer 2 Report

the methodology for chapter 2.3.2 should be better described

2.3.4 a blank test should be defined for the vis method

the FT-IR method should be supplemented in what kind of KBr 1: 100 pellets were made?

2.4. this title is too general. The work examines the dynamic viscosity and the viscosity curves in the gradient of temperature (20-80 degrees Celcujsza) and time. The title should be better chosen

Author Response

Response to Reviewer 2 Comments

Point 1: the methodology for chapter 2.3.2 should be better described

Response: According to your suggestion, the description of 2.3.2 has been expanded in revised manuscript (Line 121-125).

Point 2: 2.3.4 a blank test should be defined for the vis method

Response: We have added the blank control for the UV-vis spectroscopy analysis in revised manuscript (Line 142-143). The distilled water was used as the blank control.

Point 3: the FT-IR method should be supplemented in what kind of KBr 1: 100 pellets were made?

Response: The FT-IR method has been supplemented according to the reviewer’s suggestion. Mix together dried EPS-1 samples (1.0 mg) with potassium bromide (KBr, 100.0 mg) and press into transparent flakes. And then FT-IR spectrum of EPS-1 was collected in the wave-length range of 4000-400 cm-1 on a FT-IR spectrophotometer (BrukerCo., Ettlingen, Germany).

Point 4: 2.4. this title is too general. The work examines the dynamic viscosity and the viscosity curves in the gradient of temperature (20-80 degrees Celcujsza) and time. The title should be better chosen

Response: Thank you for your advice, we named the general title just because the content of this part of research is relatively detailed. We believe that changing the title would be unnecessary.